# Cutaneous anthrax outbreak associated with use of cattle hides and handling carcasses, Amudat District, Uganda, 2023–2024

Patrick Kwizera[ORCID]*, Richard Migisha, Hannington Katumba, Esther Nabatta, Samuel Gidudu[ORCID], Benon Kwesiga[ORCID], Job Morukileng, Lilian Bulage, Alex Riolexus Ario

Uganda Public Health Fellowship Program, Uganda National Institute of Public Health, Kampala, Uganda

* pkwizera@uniph.go.ug

## Abstract

### Background

Anthrax is a zoonotic disease that remains endemic in Uganda, particularly in cattle-keeping areas. On December 28, 2023, the first suspected human case of anthrax was detected in Amudat District. We investigated to determine the outbreak's magnitude, identify risk factors, and recommend prevention and control measures.

### Methods

We defined a suspected cutaneous anthrax case as acute onset of ≥2 of the following: skin lesions (papule, vesicle, or eschar) on exposed areas such as the hands, forearms, shoulders, back, thighs or face, localized itching, redness, swelling, or regional lymphadenopathy, in Amudat residents from December 2023–June 2024. A confirmed case was a suspected case with PCR-positive test for *Bacillus anthracis*. In unmatched case-control study (1:3 ratio), we compared exposures among 40 cases and 120 controls. We identified cases through house-to-house search, medical record reviews, and snowballing among case-persons. Human and animal samples were collected and tested, alongside an environmental assessment. We used multivariable logistic regression to identify associated risk factors.

### Results

We identified 102 cutaneous anthrax cases, including 7 confirmed cases; none died. The outbreak lasted 7 months, peaking in March 2024, with an overall attack rate of 169/100,000 (males: 196/100,000; females: 138/100,000). Use of cattle hides as bedding (OR=12; 95% CI:2.7–52) and butchering cattle carcasses (OR=6; 95% CI:1.8–19) were significantly associated with anthrax. The highest infection risk was observed among individuals with multiple exposures: butchered only (OR = 6.9, 95% CI:2.6–18), butchered and carried cattle parts (OR = 11, 95% CI:1.2–96), butchered

**Data availability statement:** The datasets upon which our findings are based belong to the Uganda Public Health Fellowship Program. For confidentiality reasons, the datasets are not publicly available. The datasets can be availed upon reasonable request from the responsible officer with permission from the Uganda Public Health Fellowship Program. Request can be directed at: it@uniph.go.ug.

**Funding:** The author(s) received no specific funding for this work.

**Competing interests:** The authors have declared that no competing interests exist.

**Abbreviations:** AR, attack rate; aOR, adjusted odds ratio; CDC, Centers for Disease Control and Prevention; CI, confidence interval; cOR, Crude Odds Ratio; VHTs, Village Health Teams; UBOS, Uganda Bureau of Statistics; UVRI, Uganda Virus Research Institute; NADDEC, National Animal Disease Diagnostics and Epidemiology Centre; WHO, World Health Organization; MoH, Ministry of Health.

and skinned (OR = 14, 95% CI:3.5–56), and butchered, carried, and skinned (OR = 17, 95% CI:1.6–219). No livestock had been vaccinated prior to the outbreak.

## Conclusion

The outbreak was associated to use of cattle hides as bedding and the butchering of cattle carcasses. We recommended community education, livestock vaccination, and safe carcass handling to prevent future outbreaks.

## Background

Anthrax is a zoonotic disease caused by *Bacillus anthracis*, a gram-positive, aerobic, spore-forming bacterium that affects both animals and humans [1]. Globally, anthrax continues to pose a significant public health threat, especially in underserved rural regions across Africa and Asia, where an estimated 20,000–100,000 cases occur annually [2,3].The World Health Organization (WHO) recognizes anthrax as one of the neglected zoonotic diseases due to its persistent burden on health and livelihoods in marginalized communities [2].

Livestock typically become infected through ingestion of contaminated feeds or water, inhalation of spores while grazing on land harboring dormant spores, which then germinate into active bacteria within the host [4]. Humans primarily acquire the infection through contact with infected animals or their products, such as meat, hides, skin, and bones [5,6]. The disease manifests in three primary forms in humans: cutaneous, gastrointestinal and inhalational, each associated with a distinct route of exposure and varying incubation periods: cutaneous (2–7 days), inhalational (1–6 days), and gastrointestinal (1–6 days) [7]. Cutaneous anthrax is the most frequently reported form, accounting for approximately 95% of human cases and may lead to fatality rates as high as 20% if not treated [2,8]. Pastoralist and agro-pastoralist communities are particularly at risk due to frequent human-animal interactions and reliance on livestock for livelihood.

In Uganda, close human-animal interactions, particularly in pastoralist regions, increase the risk of zoonotic disease transmission [9]. Between January 2017 and April 2023, the country documented 19 anthrax outbreaks across various regions, predominantly within the cattle corridor that spans the north, east, and west [10–13]. According to 2018 surveillance data, there were186 human anthrax cases and 721 livestock deaths attributed to the disease [14,15].

On December 28, 2023, a 10-year-old child from Kakworobu Village, a rural community in Amudat District, sought medical attention at a local health facility with symptoms consistent with cutaneous anthrax: an itchy lesion on the shoulder that progressed to a black eschar, accompanied by generalized body weakness. This health facility was the first to identify suspected cases of anthrax, signaling the onset of a potential outbreak. Sporadic cases continued to appear into early March 2024, prompting health authorities to alert district officials on March 5, 2024. Laboratory tests confirmed anthrax in human samples on March 25, 2024.This marked

the first documented anthrax outbreak in the district, highlighting the urgent need for a public health response. We investigated to assess the outbreak's magnitude, identify associated risk factors, and recommend control and prevention measures.

## Methods

### Outbreak area

Amudat District is located in north-eastern Uganda, within the Karamoja sub-region. shares borders with Kenya to the east, Moroto District to the north, Nakapiripirit to the west, and Kween and Bukwo District to the south. The district has a population of approximately 157,800 according to Uganda Bureau of Statistics (UBOS) and people are predominantly nomadic pastoralists. Amudat District is administratively divided into 10 lower local government units, consisting of 2 urban town councils and 8 rural sub-counties. The town councils are: Amudat Town Council and Karita Town Council. The sub-counties are: Loroo, Abiliyep, Katabok, Kangorok, Lokales, Losidok, Achorichori, and Karita (**Fig 1**).

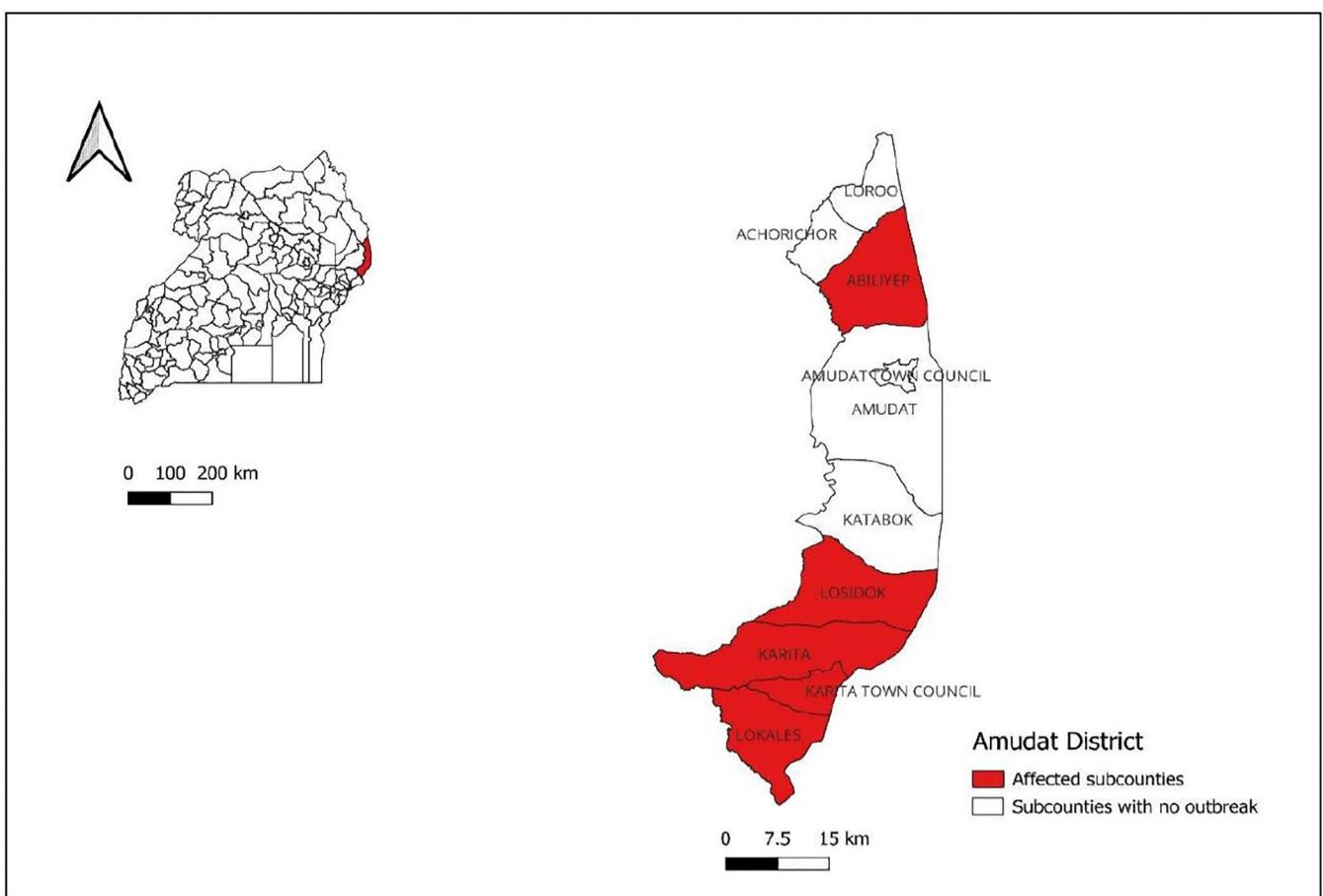

**Fig 1. Location of Amudat District in Uganda (map drawn using QGIS browser 3.10.2).**

## Case definition and finding

We defined a suspected cutaneous anthrax case as acute onset of ≥2 of the following: skin lesions (papule, vesicle, or eschar) on exposed areas such as the hands, forearms, shoulders, back, thighs or face, localized itching, redness, swelling, or regional lymphadenopathy, in Amudat residents from December 2023–June 2024.

A confirmed case was a suspected case testing PCR-positive for *Bacillus anthracis.*

To identify cases, we reviewed health facility records at health facilities serving the affected community and conducted active community case searches in collaboration with the community health workers.

## Descriptive epidemiology

We performed descriptive epidemiology on the line-listed case-patients. Case-patients were described by time, place, person characteristics. We constructed an epidemic curve to depict the distribution of cases over time. Attack rates were calculated as the number of new anthrax cases during the outbreak period divided by the total population at risk, multiplied by 100,000 using statistics based on district population estimates obtained from the District Biostatistician and stratified by age group, sex, and sub-county. Choropleth maps were developed to illustrate the geographic distribution of cases across sub-counties.

## Laboratory investigations

Samples were collected from humans, animals, and the environment during the outbreak investigation. For human cases, swabs were obtained from the active edges of cutaneous lesions (eschars) using sterile swabs. From cattle carcasses, tissue samples (ear notch or muscle), and swabs of unclotted blood oozing from natural body orifices (nose, eyes, anus) were collected where available, following strict biosafety precautions. Soil samples were also obtained from sites where cattle had died suddenly. All samples were placed in sterile containers, properly labeled, and maintained on ice during transport.

Human samples were triple-packaged and transported through the national specimen hub system to the Uganda Virus Research Institute (UVRI) laboratory in Arua District, while animal and environmental samples were sent to the National Animal Disease Diagnostics and Epidemiology Centre (NADDEC) laboratory in Entebbe, Uganda. At both laboratories, samples were processed using standard real-time PCR protocols for the detection of *Bacillus anthracis* DNA, following WHO guidelines for anthrax confirmation. Positive and negative controls were included in all assays to ensure accuracy. Laboratory staff adhered to biosafety level 2 (BSL-2) procedures throughout sample handling and analysis.

## Environmental investigations

We inspected grazing sites and areas where cattle had died suddenly, focusing on carcass disposal practices to identify potential contamination sources. Soil samples were collected specifically from these sites where cattle had died, as they were considered high-risk for environmental contamination with *Bacillus anthracis* spores and were taken to the laboratory for analysis.

Additionally, we assessed the livestock vaccination status within the affected sub-counties.

## Hypothesis generation

To generate hypotheses, we conducted face-to-face, structured interviews with 40 case-patients using a pre-tested questionnaire to identify various exposures associated with contracting anthrax. These exposures included: butchering of carcasses, use of cattle hides as bedding, and digging up buried animal remains during farming.

## Case-control study

We carried out an unmatched case-control study to identify risk factors for anthrax. A total of 40 case-patients, including all confirmed and suspected cases who could be located, were enrolled from the 102 cases identified during the outbreak.

For each enrolled case, three controls were selected randomly from the nearest households, resulting in a 1:3 case-to-control ratio. Controls were screened to ensure they had no clinical signs or symptoms of anthrax before enrolment. The remaining 62 case-patients were not included due to the nomadic nature of the Pokot population, which made it challenging to locate and follow up on some case-patients, especially those who had migrated or were unreachable during the study period. Using structured questionnaires, we collected demographic characteristics (age, sex, and sub-county) and potential exposures (butchering of carcasses from cattle that had died suddenly, skinning carcasses, carrying the meat of cattle that died suddenly, use of cattle hides as bedding from carcasses of cattle that died suddenly, and digging out animal remains).

Data were analyzed using Epi Info software. Crude odds ratios (OR) were computed at the bivariate analysis. Factors found significant at this level were included in a multivariable logistic regression model to calculate adjusted odds ratios (aOR), with significance set at $p < 0.05$. A common reference group analysis was also conducted to evaluate risk associated with multiple exposure combinations [16].

### Ethical considerations

This outbreak investigation was in response to a public health emergency and was therefore determined to be non-research. The Ministry of Health (MoH) gave permission to investigate this outbreak. In agreement with the International Guidelines for Ethical Review of Epidemiological Studies by the Council for International Organizations of Medical Sciences (1991) and the Office of the Associate Director for Science, US CDC/Uganda, it was determined that this activity was not human subject research and that its primary intent was public health practice or disease control activity (specifically, epidemic or endemic disease control activity). This activity was reviewed by the US CDC and was conducted consistent with applicable federal law and CDC policy. §§See, e.g., 45 C.F.R. part 46, 21 C.F.R. part 56; 42 U.S.C. §241(d); 5 U.S.C. §552a; 44 U.S.C. §3501 et seq. All experimental protocols were approved by the US CDC human subjects review board (The National Institute for Occupational Safety and Health Institutional Review Board) and the Uganda MoH and were performed in accordance with the Declaration of Helsinki. Permission to conduct the outbreak response was also granted by Amudat District Local Government. Prior to data collection, informed consent was obtained from all the participants who were aged 18 years or older (legal age in Uganda). For those below 18 years, consent was sought from their parents/guardians and assent was also obtained from them to participate in the study.

## Results

### Descriptive epidemiology

We identified a total of 102 case-patients during the outbreak. Forty were investigated and interviewed, of whom seven were confirmed by PCR as cutaneous anthrax and 33 were suspected cases. The overall attack rate was 169 per 100,000 population, with all cases presenting in the cutaneous form.

The mean age of the 40 cases was 24 years (range: 7 months–70 years), with the cases aged 40 years and above being the most affected (attack rate: 229/100,000), and the under 5 years group the least affected (attack rate: 88/100,000). Males had a higher attack rate (196/100,000) compared to females (138/100,000). Geographically, Losidok Sub-county recorded the highest attack rate (452/100,000), whereas Abiliyep Sub-county had the lowest (8/100,000) (**Table 1**).

Among 40 case-patients, skin itching (75%), skin swelling (73%) and eschar (70%) were the most common signs and symptoms of illness. (**Fig 2**).

On December 22, 2023, cow A died in neighboring Kween District. Its carcass was then butchered, and its meat transported to Amudat; two days later, the first suspected human case emerged with signs and symptoms suggestive of cutaneous anthrax including skin itching on the shoulders, general body weakness, and eschars on the hands and visited Karita HCIV in Karita Sub-county on December 28, 2023. On January 20, 2024, cow B died in Karita Sub-county, and

**Table 1.** Attack rates by age, sex, sub-county among case-patients during a cutaneous anthrax outbreak, Amudat District, December 2023–June 2024.

| Variables | Cases | Population | AR/100,000 |
|---|---|---|---|
| **Age(years)** | | | |
| <5 | 12 | 13,621 | 88 |
| 5-9 | 18 | 9,140 | 197 |
| 10-19 | 27 | 16,768 | 161 |
| 20-39 | 30 | 14,407 | 208 |
| ≥40 | 15 | 6,538 | 229 |
| **Sex** | | | |
| Male | 63 | 32,137 | 196 |
| Female | 39 | 28,337 | 138 |
| **Sub-county** | | | |
| Losidok | 49 | 10,846 | 452 |
| Karita | 41 | 19,284 | 213 |
| Lokales | 10 | 14,256 | 70 |
| Karita TC | 1 | 4,097 | 24 |
| Abiliyep | 1 | 11,991 | 8 |
| **Total** | **102** | **60,474** | **169** |

**AR:** Attack rate; **TC:** Town Council

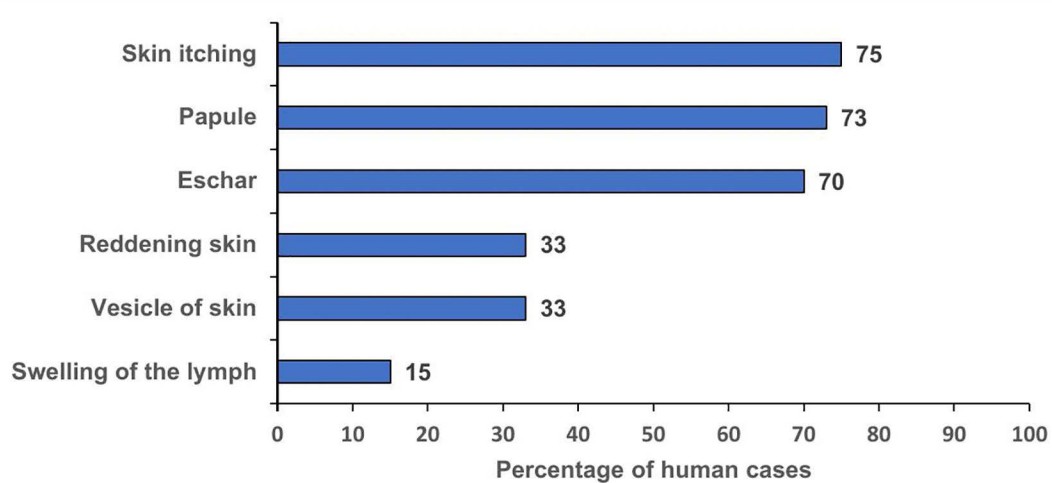

**Fig 2.** Distribution of clinical symptoms of anthrax case-patients during an anthrax outbreak, Amudat District, December 2023–June 2024 (n=40).

shortly thereafter, additional human cases emerged. Towards the end of February 2024, over 100 cattle died, this was the time cattle were returning from Kween District and it was followed by marked increase in human cases. On March 3, 2024, the index case was identified, prompting an alert to the district, on March 4, 2024, the Moroto Regional Emergency Operations Centre (REOC) was notified. On March 15, 2024, a joint team from the district and REOC verified the alerts at Karita HC4. It took 66 days to detect the outbreak, one day to be notified and 12 days for the district to initiate response.

The cases were initially confined to Karita Sub-county but later spread to Losidok and Lokales Sub-counties. We observed that during the outbreak, cattle frequently moved across sub-counties and district borders in search of pasture.

A noticeable surge that occurred in March 2024, prompted the district to respond on March 15,2024. The carcasses of infected cattle were safely disposed of to prevent further spread, and awareness campaigns were conducted in the affected sub-counties to help reduce transmission (**Fig 3**).

## Laboratory findings

Among 16 human skin lesion swabs collected, seven (44%) tested positive for *B. anthracis* by PCR at UVRI. The remaining 9 samples were negative. All six samples from the cattle carcasses and two soil samples tested negative at NADDEC.

## Environmental assessment findings

In Losidok Sub-county, we observed scattered cattle bones, suggesting prior improper disposal of animal carcasses. In Karita, we noted discarded hides that respondents reported had been used as bedding. In one family with two affected children, the hide used as bedding had been recently obtained (approximately two months prior) from a donor returning from Kween District, an area with prior anthrax outbreaks. Additionally, none of the respondents reported vaccinating their livestock against anthrax before the outbreak.

## Hypothesis generation findings

Of the 40 participants interviewed during hypothesis generation, 33(83%) reported butchering of carcasses from cattle that had died suddenly. Additionally, 33% respondents indicated they were involved in digging up animal remains (bones) during cultivation and 30% admitted use of cattle hides as bedding. We therefore considered that the use of cattle hides as bedding, digging out animal remains and the butchering of carcasses from cattle that died suddenly could be associated with the anthrax outbreak in Amudat District. (**Table 2**).

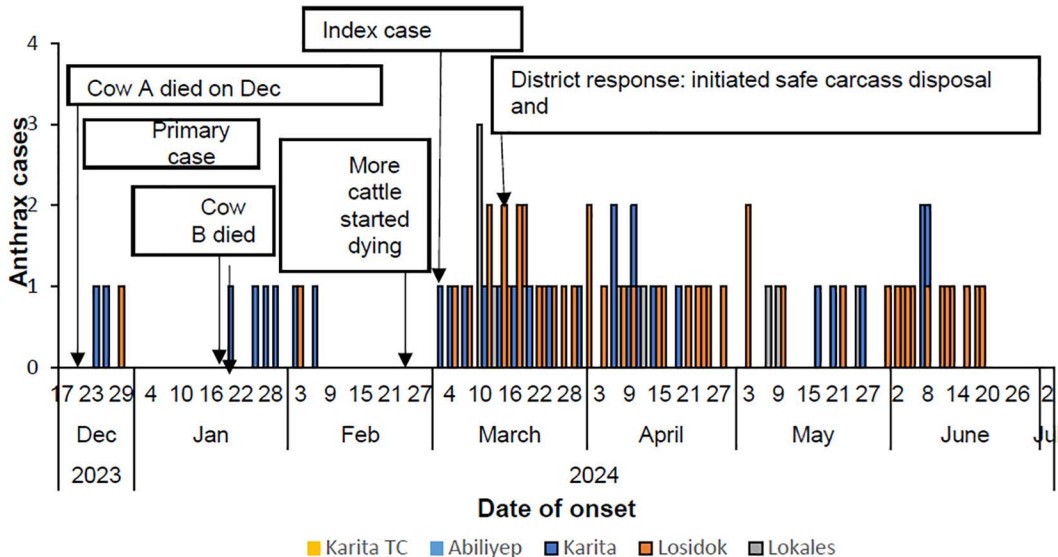

**Fig 3. Distribution of cutaneous anthrax cases across sub-counties, Amudat District, December 2023–June 2024.**

**Table 2. Hypothesis generation among case-patients during a cutaneous anthrax outbreak, Amudat District, December 2023–June 2024 (n-40).**

| Exposure | n | (%) |
|---|---|---|
| Butchering carcasses from cattle that died suddenly | 33 | (83%) |
| Dug out the animal remains during cultivation | 13 | (33%) |
| Use of cattle hides as bedding | 12 | (30%) |
| Participated in carrying of meat | 6 | (15%) |

## Case control study findings

Use of cattle hides from cattle that had died suddenly as bedding was associated with a substantially higher likelihood of cutaneous anthrax (aOR = 12, 95% CI: 2.7–52). Furthermore, individuals who butchered carcasses from cattle that died suddenly were 6 times more likely to develop cutaneous anthrax compared to those who did not (aOR = 6, 95% CI: 1.84–19) (Table 3).

Compared to the unexposed group (those with no contact with the dead cattle), individuals who engaged in more exposure activities had higher odds of contracting anthrax. Those who only butchered had an odds ratio (OR) of 6.9 (95% CI: 2.6–18). The odds were higher for those who butchered and carried cattle parts (OR=11, 95% CI: 1.2–96), butchered and skinned (OR=14, 95% CI: 3.5–56), and for those who performed all three activities butchering, carrying cattle parts, and skinning (OR=17, 95% CI: 1.6–219) (Table 4).

**Table 3. Exposure factors associated with an anthrax outbreak, Amudat District, Uganda, December 2023–June 2024.**

| Exposures | Number of participants | | cOR (95%, CI | aOR (95%, CI |
| | Cases (%) | Control (%) | | |
|---|---|---|---|---|
| **Contact with the hides** | | | | |
| Yes | 24 (60) | 32 (27) | 4.1 (1.9–8.7) | 0.8 (0.27-2.6) |
| No | 16 (40) | 88 (73) | Ref | |
| **Carrying butchered parts** | | | | |
| Yes | 6 (15) | 5 (4) | | |
| No | 34 (85) | 115 (96) | 4.1 (1.2-14) | 2.2 (0.53-9.3) |
| **Dug out animal remains** | | | | |
| Yes | 13 (32.5) | 21 (17.5) | 2.3 (1.0–5.1) | 1.1 (0.4-3.1) |
| No | 27 (67.5) | 99 (82.5) | Ref | |
| **Use of cattle hides as bedding** | | | | |
| Yes | 12 (30) | 3 (2.5) | 17 (4.0–63) | **12 (2.7–52)** |
| No | 28 (70) | 117 (97.5) | Ref | |
| **Skinning of carcasses** | | | | |
| Yes | 13(32.5) | 10 (14.4) | 5.3(2.1–13) | 1.6 (0.5-5.01) |
| No | 27(67.5) | 110 (85.6) | Ref | |
| **Butchering of carcasses** | | | | |
| Yes | 33 (82.5) | 41 (34.2) | 9.1 (3.7–22) | **6 (1.8–19)** |
| No | 7 (17.5) | 79 (65.8) | Ref | |

**Ref: reference; cOR: crude odds ratios; aOR: adjusted odds ratios; CI: confidence interval**

**Table 4. Common reference analysis of risk factors associated with cutaneous anthrax, Amudat, December 2023–June 2024.**

| Carried | Skinned | Butchered | OR | 95%CI |
|---------|---------|-----------|-----|-------|
| – | – | – | Ref | |
| – | – | + | 6.9 | 2.6-18 |
| + | – | + | 11 | 1.2-96 |
| – | + | + | 14 | 3.5-56 |
| + | + | + | 17 | 1.6-219 |

*OR: odds ratio; CI: confidence interval*

## Discussion

This outbreak represents the first documented case of anthrax in Amudat District. Our investigation identified key risk factors, including direct exposure to cattle hides and involvement in butchering activities, with a synergistic effect of these exposures amplifying the cumulative risk. Additionally, there was a lack of prior vaccination among livestock in the district. The outbreak affected five sub-counties and no human death was reported, although livestock deaths occurred in several affected areas. While the outbreak response faced significant delays in detection and response, the one-day notification requirement for the 7-1-7 metrics was met.

Using cattle hides as bedding emerged as a major risk factor for cutaneous anthrax in Amudat District, with exposed individuals more likely to develop the disease. Contaminated hides provide a plausible route for *Bacillus anthracis* transmission, consistent with previous studies from other countries [17]. To our knowledge, this is the first outbreak investigation in Uganda to suggest that using cattle hides from suddenly deceased cattle as bedding may increase the risk of cutaneous anthrax. However, the relatively small number of participants and the wide confidence intervals limit the precision of these estimates and the strength of the evidence. Future studies with larger sample sizes are needed to confirm this association and better quantify the risk. Nonetheless, these findings highlight the need for increased community awareness and the promotion of safer alternatives, with involvement of cultural and community leaders to ensure effective communication of preventive measures.

Butchering of carcasses of cattle suspected to have anthrax was also a key factor in the transmission of anthrax to humans. These findings are consistent with previous studies that have identified exposure like butchering of infected animals as a significant risk factor for human anthrax [14,18–20]. Given the risk of recurrent outbreaks in Uganda, ongoing community education on safe carcass disposal is essential to prevent future outbreaks.

The 7-1-7 metrics revealed a two-month delay in detecting and responding to the outbreak, illustrating challenges in early recognition of anthrax in previously unaffected district. Delays were likely exacerbated by limited awareness of the disease among health care providers. This delay mirrors challenges faced globally in managing public health threats [21,22]. To improve global health security, it is essential to strengthen early detection systems, enhance surveillance, train local health workers, and ensure better communication and coordination among stakeholders for faster, more effective responses to future outbreaks.

These findings have important public health implications. Targeted interventions in nomadic pastoral communities should address behaviors that increase risk, including using cattle hides as bedding and butchering of cattle carcasses. Health education campaigns should engage community and cultural leaders to promote behavioral change and safer practices. To reduce livestock-to-human transmission, routine vaccination of animals should be prioritized, as none of the livestock in the affected district had received prior vaccination. Integrated human-animal surveillance under a One Health framework is essential to ensure early detection and rapid response to future zoonotic threats.

One key limitation of this study is the likely underreporting of gastrointestinal (GI) anthrax cases, as symptoms may mimic other common GI conditions, leading to misdiagnosis. This, combined with the unfamiliarity of local healthcare providers with anthrax, given its novelty in the district likely delayed recognition and reporting, resulting in an underestimation of the outbreak's true magnitude. Additionally, some exposures were self-reported and involved a small number of cases, which may introduce recall bias. Nevertheless, these findings provide important insight into key risk factors for cutaneous anthrax in this population.

## Conclusion

This outbreak of cutaneous anthrax was primarily associated to use of cattle hides as bedding and butchering of carcasses from cattle that had died suddenly. The response was hindered by delayed outbreak detection. Prevention strategies should focus on community education, routine livestock vaccination, and safe handling and disposal of carcasses. Strengthening integrated surveillance systems across human and animal health sectors, in line with the One Health framework, could improve early detection and response to future outbreaks.

## Acknowledgments

We extend our appreciation to the District Health Team of Amudat, particularly the District Surveillance Focal Person and the surveillance team for their overall coordination and leadership during the investigation in the district. We also thank Karita Health Centre IV for their coordination with the Village Health Teams, and acknowledge the Village Health Teams for their active participation in the case search within the community. Lastly, we recognize the Uganda Public Health Fellowship Program for providing technical oversight, coordination, and throughout the investigation.

## Author contributions

**Conceptualization:** Patrick Kwizera, Richard Migisha, Hannington Katumba, Esther Nabatta, Samuel Gidudu, Job Morukileng, Alex Riolexus Ario.

**Data curation:** Patrick Kwizera, Hannington Katumba, Esther Nabatta, Job Morukileng.

**Formal analysis:** Patrick Kwizera, Job Morukileng.

**Resources:** Alex Riolexus Ario.

**Supervision:** Richard Migisha, Samuel Gidudu, Benon Kwesiga, Alex Riolexus Ario.

**Writing – original draft:** Patrick Kwizera.

**Writing – review & editing:** Patrick Kwizera, Richard Migisha, Esther Nabatta, Samuel Gidudu, Benon Kwesiga, Lilian Bulage.

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
