## [Decision Letter · Decision Letter 0]

23 Aug 2025

Dear Dr. Kwizera,

We look forward to receiving your revised manuscript.

Kind regards,

Chisoni Mumba

Academic Editor

PLOS ONE

Journal Requirements:

3. In this instance it seems there may be acceptable restrictions in place that prevent the public sharing of your minimal data. However, in line with our goal of ensuring long-term data availability to all interested researchers, PLOS’ Data Policy states that authors cannot be the sole named individuals responsible for ensuring data access (http://journals.plos.org/plosone/s/data-availability#loc-acceptable-data-sharing-methods).

4. We note that Figure 1 in your submission contain map images which may be copyrighted. All PLOS content is published under the Creative Commons Attribution License (CC BY 4.0), which means that the manuscript, images, and Supporting Information files will be freely available online, and any third party is permitted to access, download, copy, distribute, and use these materials in any way, even commercially, with proper attribution. For these reasons, we cannot publish previously copyrighted maps or satellite images created using proprietary data, such as Google software (Google Maps, Street View, and Earth). For more information, see our copyright guidelines: http://journals.plos.org/plosone/s/licenses-and-copyright.

Additional Editor Comments:

We invite you to revise your manuscript to address the reviewers’ comments in full. We believe your work has the potential to make an important contribution, but substantial revisions are needed before it can be considered further. Please attend to the following concerns.

-Provide a detailed and transparent description of your methods.

-Ensure all laboratory results are reported clearly and consistently.

-Reorganize the background and discussion for better logical flow and clarity.

-Revise terminology and phrasing throughout the text for scientific accuracy.

-Engage professional language editing support if possible.

Reviewers' comments:

Reviewer's Responses to Questions

**Comments to the Author**

1. Is the manuscript technically sound, and do the data support the conclusions?

Reviewer #1: No

Reviewer #2: Partly

Reviewer #3: Yes

2. Has the statistical analysis been performed appropriately and rigorously?

Reviewer #1: No

Reviewer #2: Yes

Reviewer #3: No

3. Have the authors made all data underlying the findings in their manuscript fully available?

Reviewer #1: No

Reviewer #2: No

Reviewer #3: No

4. Is the manuscript presented in an intelligible fashion and written in standard English?

Reviewer #1: No

Reviewer #2: No

Reviewer #3: Yes

Reviewer #1: General comment

The authors present a timely and important investigation into a cutaneous anthrax outbreak in Amudat District, Uganda, spanning December 2023 to June 2024. The topic is undoubtedly of interest and important. However, the manuscript in its current form suffers from several critical issues that substantially limit its clarity, and coherence. The structure of the paper appears overly simplistic and lacks a logical flow, making it difficult for readers to follow the progression of the investigation and its findings. Methodological details are insufficiently described, raising concerns about the robustness of the study. In addition, the presentation of data is inconsistent and at times confusing. The manuscript also contains numerous grammatical and syntactical errors that detract from its readability and professional tone. These language issues, combined with the structural and methodological shortcomings, create considerable confusion for the reader and obscure the significance of the reported outbreak.

Specific comments

Title

The title is quite lengthy and includes multiple elements. This can dilute the focus and make it harder for readers to grasp the core message quickly, and unnecessary inclusion of dates in the title.

Abstract

The Abstract is too simplistic. Although it is somewhat informative but has several flaws ranging from formatting to scientific precision and lack of Clarity in the Study Design.

Ambiguity in Case Numbers: “102 suspected and 7 confirmed cases” is clear, but it’s not stated whether confirmed cases are included within the 102 or in addition to them.

The OR suggests a wide confidence interval, indicating statistical problem or sample size issues. Therefore, results should be interpreted with caution. The conclusion restates findings but lacks nuance.

L31 Sleeping on hides can be changed to use of animal hides as bedding

Background

This background section has some useful epidemiological detail and provides a foundation for the study, but several flaws ranging from structural, scientific and stylistic flaws undermine its clarity and impact. For example, the transition from global context to Uganda-specific data to the index case is abrupt. A smoother narrative arc would help: Global → Regional → National → Local (Amudat).

L62-63 “Surveillance data from this period reported 186 human and 721 livestock death”. It’s unclear whether these were confirmed anthrax deaths, suspected, or estimated.

L68 double commas

Methods

L103 authors mention about testing samples from human skin lesions, cattle carcasses and soil. However, no such results exist in the results section, are they not important?

L98-100 why authors do not show the method or formula used for the attack rate calculation?

L117 what type of interviews?

L119 slaughtering cattle that had died suddenly? How possible is this?

L119 change sleeping on hides to use of animal hides as bedding

L122 what does 1:3 ratio mean?

L124 which ones are the rest of the case-patients? How many?

L131 again this statement: slaughtering cattle that had died suddenly?

Results

L160 40 of whom were confirmed what?

L202-203 even though authors state that 7 human samples tested positive, actual results not shown

L202-203 is this how PCR results are reported?

L213-214 Grammatical errors

L213 Slaughtering cattle that had died suddenly?

L215 change sleeping on cattle hides with use of cattle hides as bedding

L216 Slaughtering the animals that died suddenly?

L222 language

L223-224 Very wide CI for OR, Issues with modeling and sample size, results must be interpreted with caution, recall bias?

L224 again slaughter of cattle that died suddenly? How possible is this?

L234 slaughtered what?

L174 signs and symptoms? Which ones are signs and which ones are symptoms?

L178-197 does this section describe results? Is it misplaced?

L178-179 again this: cow A died and then slaughtered, what does this mean?

L180 signs and symptoms unclear?

Discussion

The text require improvement in terms of grammatical accuracy, word choice, analytical depth and redundancy. For example; (i) the discussion repeatedly emphasizes the role of cattle hides and slaughtering without synthesizing these points into a cohesive narrative. (ii) Sleeping on cattle hides” is mentioned multiple times across different paragraphs, which could be streamlined. (iii) The discussion jumps between findings, implications, and recommendations without clear transitions. (iv) The claim that this is the “first documented outbreak in Uganda where sleeping on hides…” is made several times, with slightly different wording should be improved to state the novelty once, clearly and confidently, and support it with citations or comparative context. Additional references are needed

Reviewer #2: The study is very important in highlighting inadequacies in public awareness and the consequential effects of overlooking this important activity. However, the important findings of this study have not been well presented to their expected impact. The authors keep insisting that anthrax could have spread to people who "slaughtered cattle that died suddenly". There is no such thing as slaughtering cattle that died suddenly. It is therefore imperative that this classification is removed, or given the appropriate description if the manuscript is to be accepted for publication. The methods are not well described to show exactly what was done to infer the conclusions reached. it is also necessary to describe these methods in detail. There is a lot of grammatical errors which need correction.

Reviewer #3: The work addresses a significant public health problem that is mostly neglected in most resource limited settings. The authors are commended for triangulating the data collection methods ( i.e laboratory confirmation, questionnaire, and environmental assessments) to concretize the evidence. However the following are worth noting;

1. The document has a few typographical errors that need editing.

2. The case definition of cutaneous anthrax must be strengthened in terms of location of skin lesions. This is important to rule out other diseases that present with skin lesions and to strengthen evidence of sleeping on contaminated mats.

3. The age categories in table 1 must be revised to provide a clearer picture of the ages affected by the disease in the study. to group children from 5 years to twenty obscures the possible risks in children that likely did not participate in the slaughter and skinning of cattle. This is also true for the elderly that are grouped within the 20 and above category. For this study, the very young and elderly were the ones who were more likely to be exposed to cutaneous anthrax as a result of sleeping on contaminated mats, therefore age categories could have been more rigorous.

4. Although the authors claim that cutaneous anthrax (line 260) was associated with sleeping on contaminated mats, such evidence is not demonstrated in table 4. In addition, there was no evidence of linking the cases to the sleeping mats, which even tested negative for anthrax. Therefore, this claim must be done with caution because studies conducted have demonstrated that cutaneous anthrax commonly results from slaughter, skinning and consumption of meat. The risk is mostly cumulative as most community members participate in the afore-mentioned activities, increasing the risk that is higher than the risk of sleeping on contaminated mats.

5. The results for the soil samples are rather silent.

Generally, the study design was appropriate although the authors could have achieved a deeper understanding of transmission dynamics in the outbreak if they had complemented their findings with qualitative interviews too.

The authors have indicated that datasets can not be made available as the results were derived from an outbreak investigation conducted by Uganda Public Health Fellowship Program..

**Do you want your identity to be public for this peer review?** For information about this choice, including consent withdrawal, please see our Privacy Policy

Reviewer #1: No

Reviewer #2: **Yes: ** DR Geoffrey Munkombwe Muuka (PhD)

Reviewer #3: **Yes: ** Doreen Chilolo Sitali

---

## [Author Response · Author response to Decision Letter 1]

5 Sep 2025

September 3, 2025

Chisoni Mumba

Academic Editor

PLOS ONE

RE: Cutaneous anthrax outbreak linked to using cattle hides as bedding and handling carcasses, Amudat District, Uganda, December 2023–June 2024

Dear, Chisoni Mumba

We sincerely thank the reviewers and the Editor for their thorough review and constructive remarks. We have carefully considered and addressed all comments in the revised manuscript.

We have carefully considered and addressed all comments in the revised manuscript. A detailed point-by-point response to the reviewers’ comments and Editor are attached, with corresponding revisions highlighted in the tracked-changes and clean versions of the manuscript.

We acknowledge the efforts of the reviewers and the Editor. For any further clarifications and information, please don’t hesitate to contact us.

Sincerely,

Patrick Kwizera (Corresponding Author)

RESPONSE TO REVIEWERS’ COMMENTS

Reviewer 1’s comments:

General comments

The authors present a timely and important investigation into a cutaneous anthrax outbreak in Amudat District, Uganda, spanning December 2023 to June 2024. The topic is undoubtedly of interest and important. However, the manuscript in its current form suffers from several critical issues that substantially limit its clarity, and coherence. The structure of the paper appears overly simplistic and lacks a logical flow, making it difficult for readers to follow the progression of the investigation and its findings. Methodological details are insufficiently described, raising concerns about the robustness of the study. In addition, the presentation of data is inconsistent and at times confusing. The manuscript also contains numerous grammatical and syntactical errors that detract from its readability and professional tone. These language issues, combined with the structural and methodological shortcomings, create considerable confusion for the reader and obscure the significance of the reported outbreak.

Response

We appreciate your important observation. We have revised the manuscript extensively to improve logical flow, clarity, and coherence. The background now follows a smooth narrative from global → regional → national → local (Amudat). Methods, results, and discussion have been reorganized for clarity. Language and grammatical errors have been corrected throughout the manuscript.

Specific commitments

Comment #1

Title: The title is quite lengthy and includes multiple elements. This can dilute the focus and make it harder for readers to grasp the core message quickly, and unnecessary inclusion of dates in the title.

Response

We thank you so much for this observation. We have revised and simplified the title to: “Cutaneous anthrax outbreak linked to hides and butchering of cattle carcasses, Amudat District, Uganda, December 2023–June 2024”. To retain focus while providing essential information. Dates are retained concisely to indicate study period.

Comment #2

Abstract: The Abstract is too simplistic. Although it is somewhat informative but has several flaws ranging from formatting to scientific precision and lack of Clarity in the Study

Response

We thank you for this important observation. The Abstract has been revised to improve formatting in all sections, enhance scientific precision and increase clarity. We believe the revised version addresses the concerns raised.

Comment #3

Design: Ambiguity in Case Numbers: “102 suspected and 7 confirmed cases” is clear, but it’s not stated whether confirmed cases are included within the 102 or in addition to them.

Response

We thank the reviewer for this observation: We have clarified this in the manuscript that the 7 confirmed cases were part of the 102 total cases (7 confirmed and 95 suspected cases).

Comment #4

The OR suggests a wide confidence interval, indicating statistical problem or sample size issues. Therefore, results should be interpreted with caution. The conclusion restates findings but lacks nuance.

Response

We appreciate your observation very much. The Abstract conclusion has been revised to focus on the key findings and public health implications, while limitations such as small sample size and wide confidence intervals are discussed in detail in the Discussion section.

Comment #5

L31 Sleeping on hides can be changed to use of animal hides as bedding

Response

We thank for this suggestion. While we acknowledge that “use of animal hides” is more general, we specifically investigated exposure to cattle hides in this outbreak, as the affected community primarily used hides from cattle that had died suddenly. We therefore, changed to “use of cattle hides as bedding” throughout the manuscript (See Line 34-35)

Comment #6

This background section has some useful epidemiological detail and provides a foundation for the study, but several flaws ranging from structural, scientific and stylistic flaws undermine its clarity and impact. For example, the transition from global context to Uganda-specific data to the index case is abrupt. A smoother narrative arc would help: Global → Regional → National → Local (Amudat).

Response

We thank you for this insightful observation. We have restructured the Background section to follow a logical narrative, moving from the global epidemiology of anthrax, to regional trends in Africa, national data from Uganda, and finally to the local context of Amudat District and the index case. This revision improves the clarity, flow, and scientific impact of the Background while retaining all relevant epidemiological details.

Comment #7

L62-63 “Surveillance data from this period reported 186 human and 721 livestock death”. It’s unclear whether these were confirmed anthrax deaths, suspected, or estimated.

Response

We thank you for this comment. We have clarified in the manuscript that the surveillance data include both confirmed and suspected cases. The exact breakdown of confirmed versus suspected deaths was not available (as seen on line 67-69)

Comment #8

L68 double commas

Response

We appreciate this observation: We have removed the double commas (Line 73-74)

Comment #9

Methods: L103 authors mention about testing samples from human skin lesions, cattle carcasses and soil. However, no such results exist in the results section, are they not important?

Response

We thank you for this observation, the testing samples from human skin lesions, cattle carcasses and soil are indeed important and they appear under Laboratory findings (see line 229-231)

Comment #10

L98-100 why authors do not show the method or formula used for the attack rate calculation?

Response

We thank you for this helpful comment. In response, we have now clarified in the Methods section the formula used to calculate the attack rate. “as the number of new anthrax cases during the outbreak period divided by the total population at risk, multiplied by 100,000 population (line 107-111)

Comment #11

L117 what type of interviews?

Response

Thank you for your insightful comment: We conducted face-to-face, structured interviews with 40 case-patients using a pre-tested questionnaire to collect information on demographics and potential exposures. This clarification has been added to the manuscript (Line 140-141)

Comment #12

L119 slaughtering cattle that had died suddenly? How possible is this?

Response

We thank you for your observation: slaughtering cattle that died suddenly” has been replaced with butchering of cattle carcasses, since you don’t slaughter dead cattle (seen on line 142-143) and the entire manuscript where it is applicable

Comment #13

L119 change sleeping on hides to use of animal hides as bedding

Response

We thank you for this comment: We have replaced sleeping on the hides with use of cattle hides as bedding (Line 142) and the entire manuscript

Comment #14

L122 what does 1:3 ratio mean?

Response

We thank you for this observation. The 1:3 ratio refers to the case-control study design, where for each case-patient, three controls were selected from the same community. This design increases statistical power while maintaining feasibility in field settings (line 148-149)

Comment #15

L124 which ones are the rest of the case-patients? How many?

Response

We thank you for this observation. Of the 102 case-patients identified during the outbreak, 40 cases (all confirmed and suspected) were enrolled in the case-control study. The remaining 62 case-patients could not be included due to the nomadic nature of the Pokot population, which made it difficult to locate and follow some individuals during the study period. We have clarified this in the Methods section to explicitly state the number and reason for non-enrolment, improving transparency regarding the study population (See line 150-151)

Comment #15

L131 again this statement: slaughtering cattle that had died suddenly?

Response

We thank you for this comment: We have replaced slaughtering with butchering of cattle carcasses (Line 155) and the entire manuscript)

Comment #16

Results: L160 40 of whom were confirmed what?

Response

We appreciate this observation: To clarify, of the 102 case-patients identified, 40 were investigated and interviewed of whom 7 were confirmed by PCR as cutaneous anthrax and 33 suspected cases. This distinction has been clearly stated in the revised manuscript to improve clarity (line 184-186)

Comment #17

L202-203 even though authors state that 7 human samples tested positive, actual results not shown

Response

We thank thank you for this comment. We presented the results under the Laboratory findings section, where we reported that 16 human skin lesion swabs were collected, of which 7 (44%) tested positive for Bacillus anthracis by PCR. We have ensured that this section clearly summarizes the sample sizes and testing outcomes (Line 223-224)

Comment #18

L202-203 is this how PCR results are reported?

Response

We thank you for raising this point. The PCR results are reported as number and proportion of samples testing positive (e.g., 7 of 16 human skin lesion swabs, 44%). This format is consistent with outbreak investigation reporting in epidemiological studies, where the primary goal is to summarize laboratory confirmation alongside sample size for clarity. To enhance clarity, we have ensured that the Laboratory findings section explicitly indicates the total number of samples tested and the number positive by PCR (Line 229-231)

Comment #19

L213-214 Grammatical errors

Response

Thank you for your observation. It has been corrected (line 243-244)

Comment #20

L213 Slaughtering cattle that had died suddenly?

Response

We thank you for this comment: We have replaced slaughtering with butchering of cattle carcasses (Line 243) and the entire manuscript

Comment #21

L215 change sleeping on cattle hides with use of cattle hides as bedding

Response

We thank you for this comment: We have replaced sleeping on the hides with use of cattle hides as bedding (Line 246)

Comment #22

L216 Slaughtering the animals that died suddenly?

Response

We thank you for this comment: We have replaced slaughtering with butchering of cattle carcasses (Line 246-247)

Comment #23

L222 language

Response

We appreciate your observation: We have corrected the grammar (line 246-247)

Comment #24

L223-224 Very wide CI for OR, Issues with modelling and sample size, results must be interpreted with caution, recall bias?

Response

We thank you for this observation. Some odds ratios have wide confidence intervals due to the small sample size for certain exposures and low number of confirmed cases. Recall bias is also possible for self-reported exposures. These limitations have been acknowledged in the Discussion section (Line 253-256)

Comment #25

L224 again slaughter of cattle that died suddenly? How possible is this?

Response

We thank you for your observation: slaughtering cattle that died suddenly” has been replaced with butchering of cattle carcasses, since you don’t slaughter dead cattle (seen on line 255)

Comment #26

L234 slaughtered what?

Response

We thank you for this, slaughtered dead cattle but it has been replaced with butchering of cattle carcasses (line 266)

Comment #27

L174 signs and symptoms? Which ones are signs and which ones are symptoms?

Response

We thank you for this comment, Skin itching was reported as a symptom by patients, while papules, eschar, skin redness, vesicles, and swollen lymph nodes were observed as clinical signs (Line 199-200)

Comment #28

L178-197 does this section describe results? Is it misplaced?

Response

Thank you for the comment. We note that this section describes distribution of human cases in relation to cattle deaths and movements, which is presented as the epidemic curve. We think it is best suited under results section (line 204-224)

Comment #29

L178-179 again this: cow A died and then slaughtered, what does this mean?

Response:

We thank you for this observation, cow A died and it was butchered since we don’t slaughter dead cattle (Line 204-224)

Comment #30

L180 signs and symptoms unclear?

Response:

We thank you for the observation, we have included the signs and symptoms for clarity (Line 206-207)

Comment #31

Discussion

The text require improvement in terms of grammatical accuracy, word choice, analytical depth and redundancy. For example; (I) the discussion repeatedly emphasizes the role of cattle hides and slaughtering without synthesizing these points into a cohesive narrative. (ii) Sleeping on cattle hides” is mentioned multiple times across different paragraphs, which could be streamlined. (iii) The discussion jumps between findings, implications, and recommendations without clear transitions. (iv) The claim that this is the “first documented outbreak in Uganda where sleeping on hides…” is made several times, with slightly different wording should be improved to state the novelty once, clearly and confidently, and support it with citations or comparative context. Additional references are needed

Response

Thank you for these constructive comments. We have revised the discussion to improve clarity, synthesis, and flow. References to exposures such as cattle hides and butchering of deceased cattle are now consolidated into cohesive paragraphs, with emphasis on their combined epidemiologic significance. The risk of using cattle hides as bedding is presented once, with the novelty of this finding in Uganda clearly highlighted and supported with references from other settings. The discussion has been reorganized to follow a logical structure outbreak description and findings, interpretation and risk factors, public health implications, limitations, and conclusion ensuring smooth transitions. Sentences have been refined for clarity, grammar, and conciseness. We also explicitly state the novelty of our findings and incorporated additional relevant citations to strengthen the interpretation and contextualize our results.

Reviewer# 2 comments

The study is very important in highlighting inadequacies in public awareness and the consequential effects of overlooking this important activity. However, the important findings of this study have not been well presented to their expected impact. The authors keep insisting that anthrax could have spread to people who "slaughtered cattle that died suddenly". There are no such thing as slaughtering cattle that died suddenly. It is therefore imperative that this classification is removed, or given the appropriate description if the manuscript is to be accepted for publication. The methods are not well described to show exactly what was done to infer the conclusions reached. it is also necessary to describe these methods in detail. There is a lot of grammatical errors which need correction.

Responses

We thank you for these constructive comments. We have revised the manuscript to address them as follows:

Terminology Correction

We agree with the reviewer’s observation. Throughout the manuscript, we have replaced the phrase “slaughtering cattle that died suddenly” with the more accurate description “butchering carcasses of cattle that had died suddenly”. This correction has been applied consistently in the methods, results, and discussion sections.

Presentation of Findings

The discussion has been strengthened to highlight the pub

---

## [Decision Letter · Decision Letter 1]

23 Oct 2025

Dear Dr. Kwizera,

Thank you for submitting your manuscript to PLOS ONE. After careful consideration, we feel that it has merit but does not fully meet PLOS ONE’s publication criteria as it currently stands. Therefore, we invite you to submit a revised version of the manuscript that addresses the points raised during the review process.

We look forward to receiving your revised manuscript.

Kind regards,

Chisoni Mumba

Academic Editor

PLOS ONE

Journal Requirements:

Additional Editor Comments:

Dear authors

Thank you very much for revising the paper. You have attended to most of the concerns but there are few remaining issues which are still critical to improvement of the paper.

-Your title is not concise, Please revise it. some suggestions, "Cutaneous Anthrax Outbreak Associated with Use of Cattle Hides and Carcass Handling in Amudat District, Uganda (2023-2024".

-The title and discussion still over-interpret the role of sleeping on cattle hides as a proven route of transmission, while laboratory evidence (soil = negative; hides = not tested or negative) does not confirm this pathway. The association rests solely on self-reported exposure and a small sample with wide CIs (OR = 11, 95 % CI 2.6–47). The claim of being “the first outbreak linked to use of hides as bedding in Uganda” should therefore be framed cautiously as “epidemiologically associated” rather than “linked” or “caused.” Given that the evidence is epidemiological (associative), without laboratory confirmation of the hides or hides testing positive for Bacillus anthracis, the title should reflect association, not causation.

The entire manuscript therefore must be revised to avoid causal verbs: linked to, caused by, triggered by, attributed to, etc. Use neutral epidemiologic phrasing: associated with, investigated during, epidemiologic investigation of, possible exposures during

Reviewers' comments:

Reviewer's Responses to Questions

**Comments to the Author**

Reviewer #2: (No Response)

Reviewer #3: All comments have been addressed

2. Is the manuscript technically sound, and do the data support the conclusions?

Reviewer #2: Yes

Reviewer #3: Yes

3. Has the statistical analysis been performed appropriately and rigorously?

Reviewer #2: No

Reviewer #3: Yes

4. Have the authors made all data underlying the findings in their manuscript fully available?

Reviewer #2: Yes

Reviewer #3: Yes

5. Is the manuscript presented in an intelligible fashion and written in standard English?

Reviewer #2: Yes

Reviewer #3: Yes

Reviewer #2: Most concerns have been addressed. However, there are still some issues that need to be addressed. These are highlighted in the comments on the PDF copy.

Reviewer #3: The authors have made great effort to attend to the comments by the reviewer. Stylistic and grammatical errors have been attended to. The methods and materials are now more specific and results supported by statistical analysis.

**Do you want your identity to be public for this peer review?** For information about this choice, including consent withdrawal, please see our Privacy Policy

Reviewer #2: **Yes: ** Geoffrey Munkombwe Muuka

Reviewer #3: **Yes: ** Doreen Chilolo Sitali

---

## [Author Response · Author response to Decision Letter 2]

29 Oct 2025

October 28, 2025

Prof. Chisoni Mumba

Academic Editor

PLOS ONE

RE: Cutaneous anthrax outbreak associated with use of cattle hides and handling carcasses, Amudat District, Uganda, 2023–2024

Dear Prof. Chisoni Mumba

We sincerely thank the reviewers and the Editor for their thorough review and constructive remarks. We have carefully considered and addressed all comments in the revised manuscript.

We have carefully considered and addressed all comments in the revised manuscript. Below, we provide a detailed, point-by-point response indicating how each comment was addressed in the revised version.

We acknowledge the efforts of the reviewers and the Editor. For any further clarifications and information, please don’t hesitate to contact us.

Sincerely,

Patrick Kwizera (Corresponding Author)

RESPONSE TO REVIEWERS’ COMMENTS

Editors comments:

Your title is not concise, please revise it. some suggestions, "Cutaneous Anthrax Outbreak Associated with Use of Cattle Hides and Carcass Handling in Amudat District, Uganda (2023-2024".

The title and discussion still over-interpret the role of sleeping on cattle hides as a proven route of transmission, while laboratory evidence (soil = negative; hides = not tested or negative) does not confirm this pathway. The association rests solely on self-reported exposure and a small sample with wide CIs (OR = 11, 95 % CI 2.6–47). The claim of being “the first outbreak linked to use of hides as bedding in Uganda” should therefore be framed cautiously as “epidemiologically associated” rather than “linked” or “caused.” Given that the evidence is epidemiological (associative), without laboratory confirmation of the hides or hides testing positive for Bacillus anthracis, the title should reflect association, not causation.

reported outbreak.

Response

We appreciate this important clarification. The title has been revised to: Cutaneous Anthrax Outbreak Associated with Use of Cattle Hides and Carcass Handling in Amudat District, Uganda (2023–2024) to align with your suggestion and to use epidemiologically neutral language that reflects association rather than causation throughout the manuscript

We thank you once again for your valued time you invested in reviewing our manuscript.

---

## [Editor Report · Decision Letter 2]

30 Oct 2025

Cutaneous anthrax outbreak linked to using cattle hides as bedding and handling carcasses, Amudat District, Uganda, December 2023–June 2024

PONE-D-25-41159R2

Dear Dr. Kwizera,

We’re pleased to inform you that your manuscript has been judged scientifically suitable for publication and will be formally accepted for publication once it meets all outstanding technical requirements.

Kind regards,

Chisoni Mumba

Academic Editor

PLOS ONE
---

## [Editor Report · Acceptance letter]

PONE-D-25-41159R2

PLOS ONE

Dear Dr. Kwizera,

I'm pleased to inform you that your manuscript has been deemed suitable for publication in PLOS ONE. Congratulations! Your manuscript is now being handed over to our production team.

Kind regards,

on behalf of

Dr Chisoni Mumba

Academic Editor

PLOS ONE